# Optimizing Antimicrobial Efficacy: Investigating the Impact of Zinc Oxide Nanoparticle Shape and Size

**DOI:** 10.3390/nano14070638

**Published:** 2024-04-06

**Authors:** Ana Rita Mendes, Carlos M. Granadeiro, Andreia Leite, Eulália Pereira, Paula Teixeira, Fátima Poças

**Affiliations:** 1Universidade Católica Portuguesa, CBQF—Centro de Biotecnologia e Química Fina—Laboratório Associado, Escola Superior de Biotecnologia, Rua Diogo Botelho 1327, 4169-005 Porto, Portugal; s-anrimamendes@ucp.pt (A.R.M.);; 2REQUIMTE/LAQV & Department of Chemistry and Biochemistry, Faculty of Sciences, University of Porto, 4169-007 Porto, Portugal; acleite@fc.up.pt (A.L.);; 3CINATE, Escola Superior de Biotecnologia, Universidade Católica Portuguesa, Rua Diogo Botelho 1327, 4169-005 Porto, Portugal

**Keywords:** ZnO nanoparticles, physicochemical properties, morphology and size, antimicrobial activity, food packaging

## Abstract

Zinc oxide nanoparticles (ZnO NPs) have been investigated due to their distinct properties, variety of structures and sizes, and mainly for their antimicrobial activity. They have received a positive safety evaluation from the European Food Safety Authority (EFSA) for packaging applications as transparent ultraviolet (UV) light absorbers based on the absence of significant migration of zinc oxide in particulate form. ZnO NPs with different morphologies (spherical, flower, and sheet) have been synthesized via different sol–gel methods and extensively characterized by several solid-state techniques, namely vibrational spectroscopy, powder X-ray diffraction (XRD), scanning electron microscopy/energy dispersive X-ray spectroscopy (SEM/EDS), Fourier Transform Infrared Spectroscopy (FTIR), ultraviolet–visible spectroscopy (UV-VIS), electron paramagnetic resonance (EPR), and nitrogen adsorption–desorption isotherms. The ZnO NPs were assessed for their antibacterial activity against *Escherichia coli* (gram-negative bacteria) and *Staphylococcus aureus* (gram-positive bacteria) to study the influence of morphology and size on efficacy. ZnO NPs with different morphologies and sizes demonstrated antimicrobial activity against both bacteria. The highest microbial cell reduction rate (7–8 log CFU mL^−1^ for *E. coli* and 6–7 log CFU mL^−1^ for *S. aureus*) was obtained for the sheet- and spherical-shaped NPs as a result of the high specific surface area. In fact, the higher surface areas of the sheet- and spherical-shaped nanoparticles (18.5 and 13.4 m^2^ g^−1^, respectively), compared to the flower-shaped NPs (5.3 m^2^g^−1^), seem to promote more efficient bacterial cell reduction. The spherical-shaped particles were also smaller (31 nm) compared with the flower-shaped (233 × 249 nm) ones. The flower ZnO NP resulted in a 4–5 log CFU mL^−1^ reduction for *E. coli* and 3–4 log CFU mL^−1^ reduction for *S. aureus*. The lower apparent antibacterial activity of the flower-shaped could be associated with either the lack of defects on the particle core or the shape shielding effect. Compared to *S. aureus*, *E. coli* seems to be less resistant to ZnO NPs, which may be explained by the characteristics of its cell membrane. With simple synthesis techniques, which do not allow the size and shape of the nanoparticles to be controlled simultaneously, it is a challenge to elucidate the effect of each of these two parameters on antibacterial performance.

## 1. Introduction

Nanotechnology, the science of nanomaterials, manipulates and creates nanometer-scale materials and has many applications of commercial and scientific relevance. In recent years, it has developed into a wide-ranging, multibillion-dollar global industry in multi-disciplinary areas such as engineering, physics, biological sciences, and chemistry. Numerous companies are already engaged in the production of packaging materials based on nanotechnology [1].

Nanomaterials present enhanced physicochemical characteristics compared to macroscale materials due to the very high surface area to volume ratio. As a result, nanomaterial-based food packaging presents improved mechanical, thermal and barrier properties, offering active and intelligent packaging systems that assure protection and preservation with the ability to maintain the quality and nutrition and improving the shelf-life of food [2,3,4].

Inorganic compounds at the nanoscale, such as silver, copper, titanium oxide, and zinc oxide (ZnO), present strong antibacterial activity due to their high surface area to volume ratio and unique chemical and physical properties [4]. ZnO is one such inorganic metal oxide currently found in many daily life applications, including drug delivery, cosmetics (for UV light scattering), medical installations (e.g., antibacterial paints in hospitals), dentistry (for blocking microbial leakage), and orthopedics (as a reinforcing material) [5]. ZnO is listed as a Generally Recognized as Safe (GRAS) material by the Food and Drug Administration (FDA) [6]. It has received a positive safety evaluation from the European Food Safety Authority (EFSA) for packaging applications on the basis of an absence of a significant migration in particulate forms [7]. In fact, ZnO-based composite films are promising packaging materials owing to their transparency under visible light while being able to block UV radiation. Moreover, the peculiar properties of zinc oxide nanoparticles (ZnO NPs), such as extensive surface area, biocompatibility, biodegradability, semiconductor behavior, and UV light barrier, enable their use in a vast range of applications [8].

The potential and diversity of daily life applications for ZnO NPs are due to the variety of structures and sizes of nanometric ZnO [5]. ZnO can occur in zero-, one-, two-, and three-dimensional structures. Zero-dimensional structures include nanospherical morphology [9]. One-dimensional structures make up the largest group, including nanorods [10], -needles [11], -rings [12], -tubes [13], -column [14], -belts, -wires [15], etc. The two-dimensional structures are nanopellets [16], and -plate/sheet [17]. ZnO can be obtained in three-dimensional structures such as nanoflower [18], -snowflake [19], etc. To obtain different ZnO NPs structures, several synthetic methods allow for obtaining a variety of nanoparticles with different properties, sizes, and shapes [20]. Motelica et al. [21,22] used zinc acetate dihydrate as a precursor with different alcohols to obtain ZnO NPs with different shapes and sizes.

Synthetic methods can be typically divided into three categories: chemical, biological, and physical methods. Green approaches with plant extracts and essential oils have received an exponentially growing interest for the synthesis of nanoparticles with many applications in food packaging and biomedical fields [23,24,25,26]. The use of reducing, stabilizing, and capping agents in the synthesis is very common. These agents provide stability, prevent agglomeration, and ensure uniform particle size distribution. Chemical-reducing reagents can be toxic, so using natural components as stabilizing agents can be interesting, reducing the impact of the process [27].

Foodborne illnesses are an increasingly major health problem in both developing and developed countries [28]. Moreover, pathogenic bacteria have exhibited antimicrobial resistance, and this has emerged as a hot topic of discussion among researchers in the field. Several studies have shown the interaction of ZnO NPs with foodborne pathogens [29], and ZnO NPs are currently being investigated as an antimicrobial agent in both microscale and nanoscale formulations with potential applications in food preservation. It is recognized that nanoparticles exhibit properties significantly different from microparticles [30]. Once introduced into a polymeric matrix, the interaction between the packaging and the food occurs, inhibiting microorganisms’ growth [31].

The exact mechanism of action of ZnO NPs is still not completely understood. The antimicrobial activity may be due to several mechanisms: release of antimicrobial ions [32], direct contact of ZnO NPs with the cell walls of microorganisms resulting in damage to the integrity of bacterial cells [33], and the formation of reactive oxygen species (ROS) by the effect of light radiation [34]. The microbiological effects of the ZnO NPs depend mainly on particle size, concentration, and exposure time to the bacteria cell [35]. A higher concentration of smaller particles with a larger surface area is proven to have a more efficient antibacterial behavior [36,37].

The different shapes and morphologies of ZnO NPs have also impacted antimicrobial activity. Thus, it is essential to control the experimental parameters such as solvents, precursor types, and physicochemical settings, including temperature and pH [29]. Flower-shaped ZnO NPs have revealed higher biocidal activity against *Staphylococcus aureus* and *Escherichia coli* than the spherical-shaped ZnO NPs [38]. The cuboidal-shaped nanoparticles were found to have more antibacterial activity compared to spherical- and hexagonal-shaped particles [25]. ZnO nanostructures can influence their internalization mechanism, such as rods and wires penetrating cell walls of bacteria more easily than spherical ZnO NPs [39]. Although many studies have been published, particle characterization has often not been fully reported. For this reason, the work herein described attempts to perform an extensive structural and morphological evaluation of the nanoparticles.

The present study aims to synthesize ZnO NPs with spherical, flower, and sheet morphologies and investigate the influence of morphology and size on antibacterial activity toward *E. coli* (gram-negative bacteria) and *S. aureus* (gram-positive bacteria).

Several synthetic methods were tested following previous reports in the literature, such as hydrothermal, solvothermal, microwave, and ball milling, to obtain different shapes and sizes by varying the synthesis conditions [40,41,42,43]. ZnO NP characterization was performed to assess their structural and physicochemical properties using scanning electron microscopy/energy dispersive X-ray spectroscopy (SEM/EDS), X-ray powder diffraction (XRD), Fourier-transform infrared spectroscopy (FTIR), ultraviolet–visible spectroscopy (UV-VIS), electron paramagnetic resonance (EPR), and nitrogen adsorption–desorption isotherms. Antibacterial tests were carried out using a viable cell count assay.

## 2. Materials and Methods

### 2.1. Synthesis of ZnO NPs

The synthesis of ZnO NPs involved three distinct sol–gel methods, each tailored to yield a unique shape.

Spherical shape (ZnO-SP): Zinc acetate dihydrate (2.20 g) was dissolved in 60 mL of ethanol at 60 °C for 30 min. Simultaneously, 2.52 g of oxalic acid dihydrate was dissolved in 40 mL ethanol at 50 °C. After that, the oxalic acid solution was slowly added under stirring into the warm ethanolic solution of zinc acetate. The resulting white gel was dried at 80 °C for 20 h and calcined at 500 °C for 2 h in a muffler furnace [44].

Flower shape (ZnO-FL): Zinc nitrate hexahydrate (2.97 g) was dissolved in 50 mL of distilled water at 60 °C for 15 min. Subsequently, 0.99 g of sodium hydroxide dissolved in 50 mL distilled water at 60 °C was rapidly added to the zinc metal salt solution over a 3 s time period. After 60 min of reaction time, the solution was cooled down to room temperature. The precipitate was recovered via centrifugation (4000 rpm, 5 min) and washed three times with distilled water. The ZnO NPs were dried overnight at 80 °C and were further dried in a vacuum oven at 60 °C and 200 mmHg for 2 h [45].

Sheet shape (ZnO-SH): Zinc acetate dihydrate (4.38 g) was dissolved in 100 mL of distilled water at room temperature. Meanwhile, 5.99 g of sodium hydroxide was dissolved in 50 mL of water and added dropwise to the zinc ion solution with continuous stirring. The resulting solution was placed in a sonication bath (Bandelin Sonorex) and sonicated for 1 h. The white precipitate was filtered and washed with methanol four times to eliminate ionic impurities and then dried at room temperature in a desiccator [11].

Reagents and solvents were purchased as reagent-grade and used without further purification. Zinc acetate dihydrate, zinc nitrate hexahydrate, and methanol were purchased from Sigma-Aldrich, Steinheim, Germany. Sodium hydroxide was purchased from Fisher Chemical, Brussels, Belgium. Ethanol is from Valente and Ribeiro, Belas, Portugal, and oxalic acid dihydrate is from Merck, Darmstadt, Germany.

### 2.2. Physicochemical Characterization of ZnO NPs

The characterization of synthesized ZnO NPs was carried out to determine their structural and physicochemical properties.

#### 2.2.1. Scanning Electron Microscopy and Energy Dispersive X-ray Spectroscopy

SEM was used for the morphological and size studies, and EDS was used for elemental detection analysis. The analyses were performed using a FEI QUANTA 400 FEG ESEM microscope coupled with an EDAX PEGASUS X4M (FEI, Hillsboro, OR, USA) equipment operating at an accelerating voltage of 15 and 25 kV. The ZnO NPs samples were studied as powders. The size of the ZnO NPs was determined via analysis of SEM images using ImageJ software (version 1.8.0).

#### 2.2.2. X-ray Powder Diffraction

XRD has been used to determine the crystallinity and phases of the ZnO NPs. The experiments were performed using an X’Pert MPD Philips diffractometer (Philips, Almelo, The Netherlands) equipped with an X’Celerator detector and a flat-plate sample holder in a Bragg–Brentano parafocusing optics configurations (45 kV, 40 mA) with a Cu radiation at 1.5406 Å for an angle range of 20–80° at a scan step of 0.039. The phases were identified using Highscore Plus software with the ICSD database. The data were subsequently collected and interpreted by comparing them with the Joint Committee on Powder Diffraction Standards (JCPDS) card no. 36–1451 [46].

#### 2.2.3. Fourier-Transform Infrared Spectroscopy

The functional bonds present in the ZnO NPs were studied via FTIR spectrum using a Spectrum 100 FT-IR spectrometer with a DTGS detector (Perkin Elmer, Shelton, FL, USA). Spectra were acquired in diffuse reflectance mode through a PIKE Technologies Gladi attenuated total reflectance (ATR) accessory within the wavenumber interval of 4000 to 500 cm^−1^, with a resolution of 4 cm^−1^ and 20 scans.

#### 2.2.4. Ultraviolet–Visible Spectroscopy

UV-VIS spectroscopy was carried out using a Thermo Nicolet Evolution 100 (Thermo Fisher Scientific, Waltham, MA, USA). The powdered ZnO NPs were suspended in deionized water with a concentration of 1 mg mL^−1^, and UV-VIS spectra were recorded between 300 and 600 nm of wavelength.

#### 2.2.5. Surface Area and Porosity

The specific surface area and porosity of ZnO NPs were assessed after determining the corresponding nitrogen adsorption–desorption isotherms at 77 K, obtained with a ASAP 2020 apparatus (Micromeritics, Norcross, GA, USA) using the relative pressure in the 0.03–0.2 range. The samples were previously outgassed by heating up to 120 °C under vacuum conditions. The particles’ specific surface area was determined using the Brauner–Emmet–Teller (BET) method, and the pore volume and width were determined using the Barrett–Joyner–Halenda (BJH) method [47].

#### 2.2.6. Electron Paramagnetic Resonance

EPR spectra were recorded using a ELEXSYS E500 spectrometer (Bruker, Karlsruhe, Germany) operating at 9 GHz (X-band) using the following experimental conditions: modulation frequency of 100 kHz, microwave power of 20 mW, modulation amplitude of 1 G, 60 dB of receiver gain, acquisition time of 300 s, and five scans. Solid ZnO NPs samples were placed in a capillary, then in the quartz EPR tubes and analyzed.

### 2.3. Antibacterial Activity of ZnO NPs

The antibacterial activity of the ZnO NPs against the foodborne pathogenic bacteria *S. aureus* ATCC 6538 (gram-positive) and *E. coli* ATCC 29215 (gram-negative) was evaluated for the three ZnO NPs at 22 °C. Test bacteria were aseptically inoculated into brain–heart infusion (BHI) broth and subsequently incubated at 37 °C for 16 h to obtain a bacterial population of 10^8^–10^9^ CFU mL^−1^. Cells were harvested via centrifugation at 7000 rpm/8 min and resuspended at the initial volume in Ringer’s solution. A solution of 100 mg mL^−1^ of ZnO NPs was prepared by adding the required amount of nanoparticles to the inoculum and incubated at 22 °C. Samples were collected after 4, 24, 48, and 168 h (1 week). These data points were selected based on preliminary analysis to evaluate the microorganisms’ reduction kinetics. The time was extended to verify the total reduction, taking into consideration the potential application for food storage for up to one week. The samples were plated on BHI agar plates after appropriate dilution to determine bacterial survival. Cell suspensions without ZnO NPs were used as controls. Three independent replicates were carried out. Cell reduction was determined as follows:R (t) = Log (N/N_0_) (1)
where “R (t)” stands for the cell reduction over time, “N” corresponds to the CFU mL^−1^ after exposure to NPs for a certain time, and “N_0_” denotes the initial level before exposure to NPs (CFU mL^−1^).

### 2.4. Data Handling and Statistical Analysis

Statistical analysis was performed using IBM SPSS Statistics (version 28) and GraphPad Prism (version 8.4.2). Mean and standard deviations were presented in the graphic representation. Linear mixed–effects models were used to assess differences between repeated measures of different morphologies in antimicrobial activity. One-way ANOVA and Tuckey’s post hoc analysis for pairwise comparison of more than two means were used to identify differences between each individual time point. A two-sided *p* < 0.05 was considered statistically significant.

## 3. Results and Discussion

### 3.1. Characterization of ZnO NPs

The sol–gel method was selected to prepare the nanoparticles to be used in this study because it yields considerable amounts of sample and is recognized as a simple, low-cost, reliable, and reproducible method with relatively mild synthesis conditions [20]. Other methods are not so simple and, despite presenting good results regarding the shape and size of ZnO NPs, do not allow sufficient samples to be obtained to proceed with the work.

The morphology and chemical composition of the prepared samples were assessed using SEM/EDS techniques. SEM results are presented in Figure 1a–f, where the shapes and sizes of the ZnO NPs synthesized using the different methods are evident. Spherical-shaped nanoparticles with an average diameter of 31 ± 6 nm were obtained (Figure 1a,b), where nanoparticles are clustered in filaments that grow in one direction. Flower-shaped ZnO NPs were obtained with a notorious particle individualization with a mean petal diameter of 233 ± 58 nm and length of 249 ± 74 nm (Figure 1c,d). Sheet-shaped ZnO NPs are represented in Figure 1e,f with mean averages of 243 ± 96 and 108 ± 10 nm. The size distribution was assessed through the SEM images, and detailed results are presented in Appendix A. It is observed that ZnO-FL has a wider range of particle size, while ZnO-SP and ZnO-SH exhibit a narrower size distribution. The polydispersity index was lower than 0.4 for all shapes, indicating the relative monodisperse nature of the distribution.

The EDS spectra in Appendix A confirm the purity of the ZnO NPs for all shapes obtained. In fact, only the peaks corresponding to the Zn lines at 1.0, 8.7 and 9.7 KeV, and O (lower than 1 keV) were detected. The peak corresponding to elemental carbon may be attributed to the carbon tape used to fix the sample. Samples obtained with ZnCl as a precursor [48] presented Cl in EDS spectra, and several removal techniques were tested without success. The presence of Cl is due to strong bonds between Zn and Cl. This sample was not considered for further work given the potential effect of Cl in the antimicrobial activity, interfering with pure ZnO.

The crystallinity of ZnO NPs was evaluated using powder XRD (Figure 2). The XRD patterns of all ZnO NPs indicate the presence of the main diffraction peaks at 2θ ≈ 32, 34, 36, 48, 57, 63, 66, 68, 69, 73, and 77°, which correspond to (100), (002), (101), (102), (110), (103), (200), (112), (201), (004), and (202) crystal planes of ZnO NPs. The peaks match well with the reported JCPDS card no. 36–1451, which validates the crystalline nature and the wurtzite structure of the synthesized ZnO NPs with different morphologies. No additional peaks from impurities were detected, which confirms the phase purity of the samples [49,50].

FTIR spectra (Figure 3) provide details about the chemical bonds involved in the synthesized ZnO NPs. The three ZnO shapes presented the characteristic band assigned to the Zn-O stretching mode located at 546 cm^−1^, 545 cm^−1,^ and 568 cm^−1^ for spherical, flower, and sheet shapes, respectively [51]. The sheet shape presents two bands at 1551 and 1433 cm^−1^ related to the presence of acetate ions adsorbed on the nanostructure surface. Similar results were reported for particles produced with the same precursor (zinc acetate dihydrate) [21].

ZnO NPs were approved to be used as UV absorbers after polymers were incorporated for food packaging applications, for unplasticized polymers at up to 2% by weight, and in all polyolefins [7]. Therefore, the UV-VIS barrier for the different shapes is of particular interest. Different concentrations of ZnO NPs were tested to acquire the corresponding UV-VIS spectra. The readings were taken after dispersion and a fast reading at a concentration of 1 mg mL^−1^.

Results from UV-VIS spectroscopy in the spectral range between 300 and 600 nm are presented in Figure 4. All ZnO NPs samples exhibited an absorption band at 375 nm but with different absorption intensities. ZnO-SP and ZnO-SH present similar absorption of 0.36 and 0.38, respectively. The ZnO-FL presents the highest absorption value (0.58), which the indirect effect of particle size can explain. Similar absorption peaks of ZnO NPs were obtained in other studies, for instance, at 377 nm [52], 367 nm [53], and 362 nm [49], but there are not many works focusing on the influence of different shapes/sizes of ZnO NPs on UV-VIS.

The textural properties of ZnO NPs were determined via nitrogen adsorption–desorption isotherms (Figure 5). The behavior of ZnO-SP and ZnO-SH are similar, exhibiting micropores (type II isotherm according to IUPAC classification [54]). ZnO-FL shows hysteresis, which is characteristic of mesoporous samples (type IV isotherm) [55]. The textural properties, such as specific surface area, pore width, and pore volume, were determined from the isotherm and are presented in Table 1. The sheet-shaped nanoparticle is the one that presents the highest specific surface area (18.5 m^2^ g^−1^), pore width (11.9 nm), and pore volume (0.07 cm^3^ g^−1^), while the ZnO-FL has the lowest values. Sulciute et al. [56] determined the surface area for different morphologies of ZnO NPs. In that study, the large tetrapods ZnO were revealed to have the highest surface area (22 m^2^ g^−1^), which is similar to the value herein reported for ZnO-SH.

Another study described the synthesis of ZnO NPs with the assistance of different types of surface-stabilizing agents. The highest value of the specific surface area was measured for the ZnO/polyvinyl alcohol (spherical shape) with a value of 25.70 m^2^ g^−1^ compared to the ZnO/polyvinyl pyrrolidone (hexagonal prismatic rods) and ZnO/polyglutamic acid (ellipsoid) that presented 21.74 m^2^ g^−1^ and 9.78 m^2^ g^−1^, respectively [36]. Higher values were obtained for spherical and hexagonal prismatic rods compared to this work. However, stabilizing agents were used, which may have influenced these values.

There are not many works that study the specific surface area and porosity of different shapes of ZnO NPs. Therefore, it is of interest to explore these parameters in depth. The pore volume distribution (Figure 6) varies with the shape of the ZnO NPs. The ZnO-SPs have a lower pore volume in the range of 2–3 nm with a narrow distribution focused on this size, and they present a total pore volume of 0.03 cm^3^ g^−1^. ZnO-SH presents pores within the 2–4 nm range but with a slightly more dispersed distribution compared to the spherical shape and demonstrates the highest pore volume of 0.07 cm^3^ g^−1^. ZnO-FL has pores of various sizes and is more dispersed than the other shapes, with a volume of pores of 0.02 cm^3^ g^−1^. However, higher pore volume values are observed for sizes between 6 and 10 nm, justifying the phenomenon of hysteresis (mesoporous) of the adsorption–desorption isotherms.

EPR spectroscopy was used to identify intrinsic defects in the nanostructured ZnO samples. According to the core–shell model, the EPR signal at g = 1.961 is attributed to core defects arising from negatively charged Zn vacancies acting as shallow acceptors, while the signal at g = 2.005 is attributed to a shell containing a high concentration of surface defects. These defects have been identified as positively charged oxygen vacancies acting as deep donors [57,58,59].

Figure 7 illustrates the results from continuous-wave EPR at X-band obtained for the solid ZnO samples at room temperature. Concerning the results for ZnO-SP, two signals with g values of 2.0052 and 1.9612 are discernible, with the signal attributed to core defects appearing much more intense than that attributed to surface defects, suggesting an increased number of defects in the core of the nanoparticles. Similar observations were noted for ZnO-SH nanoparticles (g values of 2.0055 and 1.9592). However, in this case, the more intense signal is associated with surface defects.

This finding aligns with earlier studies that confirmed a notable concentration of oxygen vacancies in ZnO-SH. The research conducted by Wang et al. [60] suggested that increasing the BET surface area of ZnO is an effective strategy for enhancing its surface oxygen vacancies, a hypothesis supported by BET surface area measurements performed on these particles. The outcomes of this study establish ZnO-SH as having the highest specific surface area. This accounts for the higher intensity signal observed in surface defects associated with oxygen vacancies when compared to the signal due to core defects.

Upon analyzing the EPR spectrum of ZnO-FL nanoparticles, only one signal attributed to surface defects, g values of 2.0048, can be observed, possibly due to the morphology of the nanoparticles and low BET surface area.

Based on these results, it is plausible that the flowers either lack defects in the core, or perhaps the shape of the nanoparticles shields the core defects, preventing their appearance in the EPR spectrum.

EPR measurements are scarce in the literature addressing various nanoparticle morphologies, and the few conducted did not observe differences in the EPR spectra for the tested nanoparticles. For instance, in the case of ZnO nanoflower, nano sponge, and nano urchin annealed at different temperatures [61].

### 3.2. Antibacterial Activity

Previous studies have already indicated that antimicrobial activity is influenced by the size, shape, and specific surface area of ZnO NPs [35]. However, a systematic characterization of the NPs associated with antimicrobial activity is lacking. In this work, the influence of different morphologies of ZnO NPs on the survival of *E. coli* and *S. aureus* at 22 °C was investigated (Figure 8).

When comparing the reduction in *E. coli* at 22 °C in the presence of NPs, significant differences (*p* < 0.001) were observed at 24 h among all shapes. Only ZnO-FL did not induce a reduction in *E. coli* compared to the control (*p* = 0.819). ZnO-SH demonstrated the highest antibacterial activity with counts below the detection limit of the enumeration technique after 24 h. A similar reduction was observed for bacteria in contact with the ZnO-SP, but only after one week. *E. coli* in contact with ZnO-FL showed a reduction of 4–5 log CFU mL^−1^ after one week.

Regarding *S. aureus*, significant differences (*p* < 0.001) were observed among all shapes after 24 h in contact with NPs at 22 °C. However, as observed for *E. coli*, counts of *S. aureus* were not reduced in the presence of ZnO-FL for 24 h compared with the control (*p* = 0.949). When exposed to ZnO-SP and ZnO-SH, counts of *S. aureus* were reduced to below the detection limit of the enumeration technique after one week, albeit with slightly faster reduction kinetics in the presence of ZnO-SP. Conversely, when exposed to ZnO-FL, a lower reduction of 3–4 log CFU mL^−1^ was observed after one week.

The results obtained in this study are aligned with previous studies, demonstrating that a larger surface area of NPs leads to better antibacterial activity [35]. ZnO-SH and ZnO-SP presented the highest antibacterial activity, with a total reduction of both bacteria after one week of contact, with ZnO-SH showing a relatively more pronounced faster effect on *E. coli*. The ZnO-FL presents the lowest antibacterial effect, reflecting the ZnO NPs properties. This is in line with the specific surface area results since the sheet shape presents a higher surface area (18.5 m^2^ g^−1^), followed by the spherical shape that presents a smaller size (31 nm) and a specific surface area of 13.4 m^2^ g^−1^. On the other hand, the ZnO-FL presents the highest size (249 × 233 nm) and the lowest surface area (5.3 m^2^ g^−1^), resulting in a less efficient antibacterial effect. Furthermore, the absence of core defects associated with zinc vacancies in ZnO-FL may contribute to its lower antibacterial activity. The presence of surface defects associated with oxygen vacancies in all shapes can be correlated with the antibacterial effect because the attack of reactive hydroxyl radical on the bacterial cell causes membrane damage [38]. Despite these differences, all three types of nanoparticles exhibit antibacterial activity.

Few studies have investigated the antimicrobial activity of ZnO NPs with different shapes and sizes. For example, Sharma et al. [25] used the agar well diffusion method to assess the effectiveness of cuboidal, spherical, and hexagonal-shaped ZnO NPs against *Bacillus subtilis*, *S. aureus,* and *E. coli*. They found that cuboidal-shaped ZnO NPs (40–45 nm) exhibited relatively higher antibacterial activity as compared to spherical (60–180 nm) and hexagonal-shaped (63 nm) NPs. However, the particle surface area was not determined. Motelica et al. [21] performed an agar diffusion assay with different ZnO NPs types, concluding that the size of ZnO NPs has a higher impact on gram-negative than on gram-positive bacteria. The gram-positive bacteria seem to be affected by both the size and morphology of nanoparticles.

In another study, the antibacterial activity of ZnO NPs with hexagonal, ellipsoidal, and spherical morphology was tested against *E. coli* and *S. aureus*. The highest microbial cell reduction rate was recorded for the spherical ZnO NPs [36]. This result can be attributed to the lower size (30 nm) and the higher specific surface area (25.70 m^2^ g^−1^) of the spherical NPs. These values are comparable with those found in this work despite the slightly higher specific surface area. In contrast to our findings, another study reported that flower-shaped ZnO revealed higher biocidal activity against *S. aureus* and *E. coli* compared to rod-shaped ZnO NPs. This trend was attributed to the antibacterial activity increasing with decreasing particle crystallite size or increasing surface area, following the order flower (28.8 m^2^ g^−1^) > sphere (19.2 m^2^ g^−1^) > rod (15.5 m^2^ g^−1^). However, in the present work, the ZnO-FL exhibited the lowest specific surface area, justifying its reduced antibacterial activity [38].

In this study, *E. coli* shows lower resistance to ZnO NPs compared to *S. aureus*. Similar results have been reported in previous studies, demonstrating a more pronounced inhibitory effect on gram-negative bacteria compared to gram-positive bacteria [62,63]. For instance, ZnO nanotubes with a specific surface area similar to ZnO-SH in this study (17.8 m^2^ g^−1^) were studied using the disc diffusion technique, revealing that gram-negative bacteria are less resistant in comparison to gram-positive bacteria, as they are inhibited at lower concentrations of ZnO NPs [64].

The differences observed can primarily be attributed to variations in the cell wall structure. Gram-positive bacteria typically have a thicker peptidoglycan layer (20–80 nm). In contrast, gram-negative bacteria have a thinner peptidoglycan layer (<10 nm), which is surrounded by an outer membrane containing lipopolysaccharide, phospholipids, and proteins [65]. Even with an extra layer, the cell wall provides less resistance, making these bacteria more susceptible to ZnO activity [63]. These differences in cell envelope confer different properties that influence responses to external stresses. However, it is important to note that the overall impact depends on the species/strain and characteristics that are not usually considered in the literature, making generalizations difficult.

## 4. Conclusions

Different synthetic methods for the preparation of ZnO NPs were performed to obtain nanoparticles with different morphologies. The sol–gel method was selected based on its simplicity, low cost, and reproducibility and was applied with different precursors, reductor agents, and physicochemical settings, such as temperature and pH, to obtain three different types of nanoparticles.

A full characterization of synthesized ZnO NPs was carried out to determine their structural and physicochemical properties. Particles were obtained with spherical-, flower-, and sheet-shaped ZnO NPs in high crystalline and wurtzite structures with no impurities.

The different morphologies obtained for ZnO NPs have a direct influence on the particle size, size distribution, pore volume, and specific surface area. In fact, spherical-shaped particles exhibited the smallest size (31 nm), while the sheet-shaped ZnO NPs showed the highest specific surface area (18.5 m^2^ g^−1^ against 5.3 m^2^ g^−1^ of the flower). Moreover, the spherical- and the sheet-shaped particles showed a microporous structure (2–4 nm), while flower-shaped ZnO NPs exhibited a mesoporous structure (6–10 nm) as shown by the hysteresis behavior in the nitrogen adsorption–desorption isotherms.

Both ZnO-SP and ZnO-SH present particle core and surface defects. In the case of ZnO-SH, the surface defects associated with oxygen vacancies are higher than the core defects. The ZnO-FL showed only surface defects. The lack of evidence of core defects can be due to the shielding effect of the particle shape.

ZnO NPs are used as UV absorbers after incorporation into polymers for food packaging applications, and the UV spectra exhibited a peak absorption at 375 nm with higher intensity for the flower-shaped particles.

Antibacterial studies were performed to study the influence of the ZnO NPs in gram-negative *E. coli* and gram-positive *S. aureus* at 22 °C. All three shapes presented antibacterial activity; however, there were significant differences depending on the morphology, size, and specific surface area of the ZnO particles. For *E. coli*, ZnO-SH and ZnO-SP presented the highest antibacterial activity (total reduction obtained after 24 and 168 h, respectively), with the ZnO-SH showing a relatively faster action, justified by the higher specific surface area. The ZnO-FL showed the lowest antibacterial activity. Comparing both microorganisms, *E. coli* seems to be less resistant to ZnO NPs, while *S. aureus* is more difficult to reduce, which can be explained by the cell membrane characteristics.

This work brings light on the synergetic effect of the factors influencing the antibacterial activity of ZnO NPs. In particular, specific surface area and particle core defects are key parameters that play a crucial role in the antimicrobial reduction of ZnO NPs. However, these parameters depend on the particle shape and size of the ZnO NPs, which in turn are determined via the synthetic route. It is often a challenge to obtain different particle sizes for the same shape and vice versa in order to separate and elucidate the effects of these two parameters.

## Figures and Tables

**Figure 1 nanomaterials-14-00638-f001:**
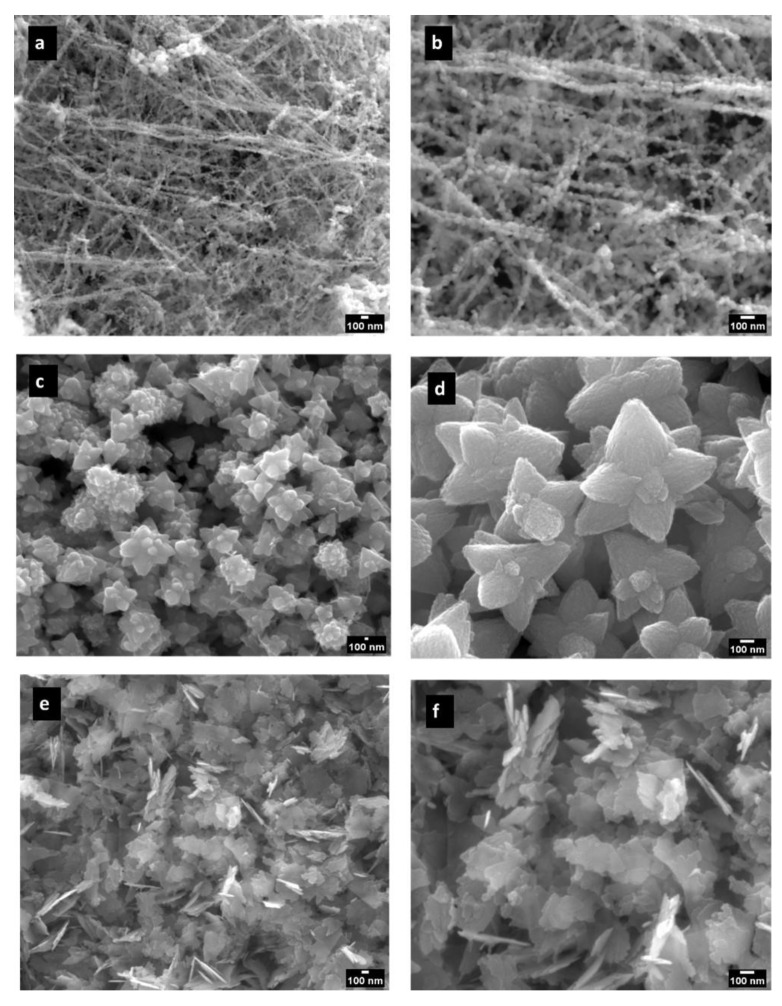
SEM micrographs of ZnO NPs: ZnO-SP (**a**,**b**); ZnO-FL (**c**,**d**); and ZnO-SH (**e**,**f**). Magnification: left side—50,000×; right side—100,000×.

**Figure 2 nanomaterials-14-00638-f002:**
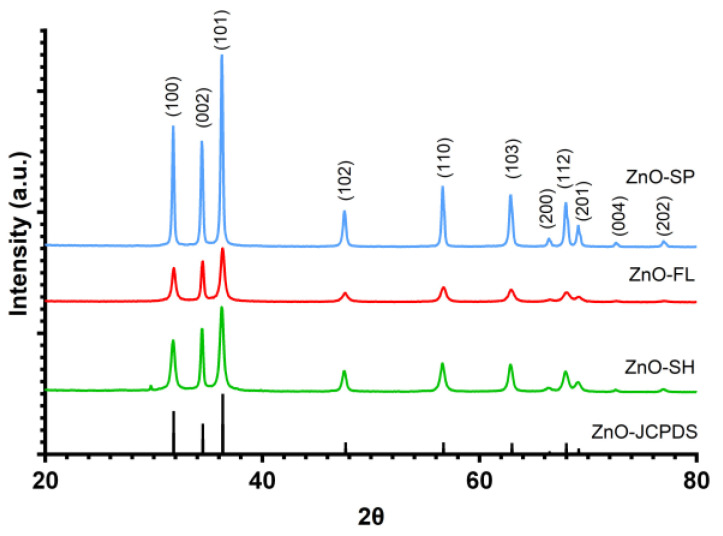
Powder XRD patterns of ZnO-SP, ZnO-FL, ZnO-SH, and JCPDS card (36–1451) of ZnO.

**Figure 3 nanomaterials-14-00638-f003:**
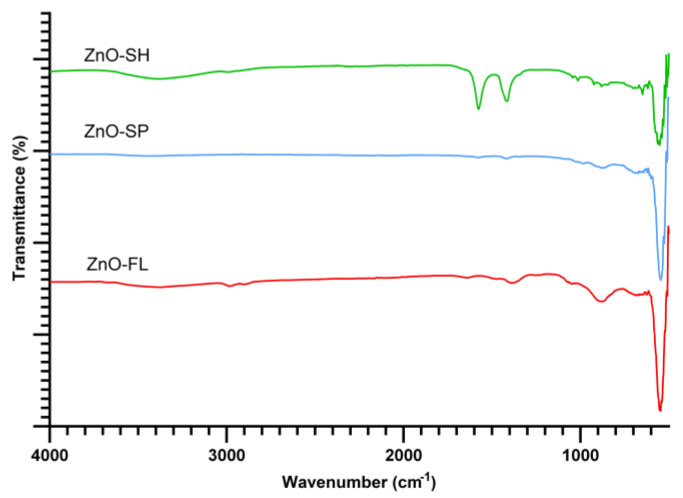
FTIR spectra of ZnO-SP, ZnO-FL, and ZnO-SH.

**Figure 4 nanomaterials-14-00638-f004:**
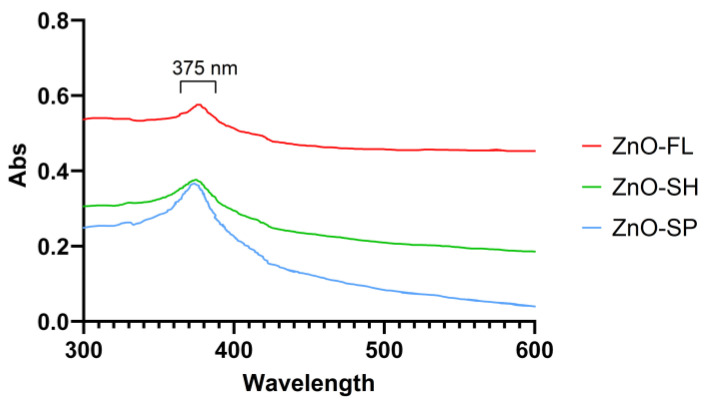
UV-VIS spectrum of ZnO-SP, ZnO-FL, and ZnO-SH.

**Figure 5 nanomaterials-14-00638-f005:**
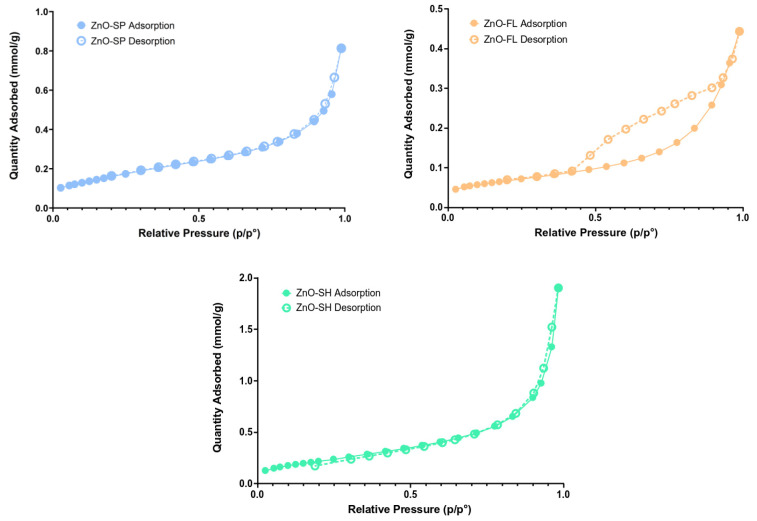
Adsorption—desorption isotherms of ZnO-SP, ZnO-FL, and ZnO-SH.

**Figure 6 nanomaterials-14-00638-f006:**
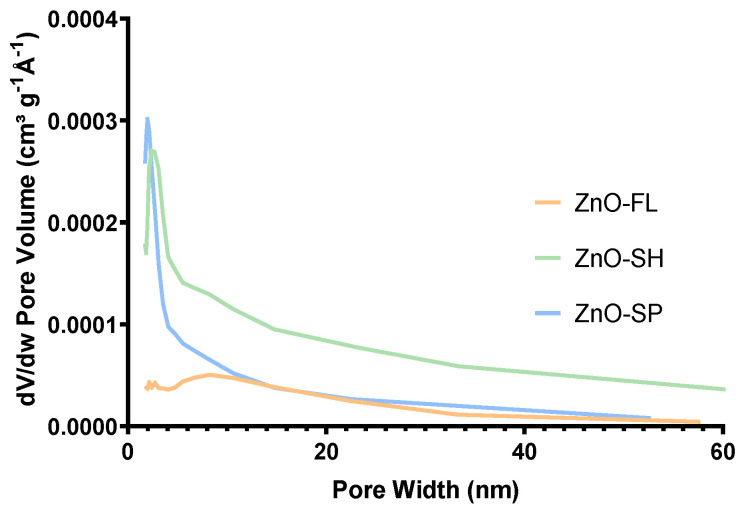
Pore volumes of ZnO-SP, ZnO-FL, and ZnO-SH.

**Figure 7 nanomaterials-14-00638-f007:**
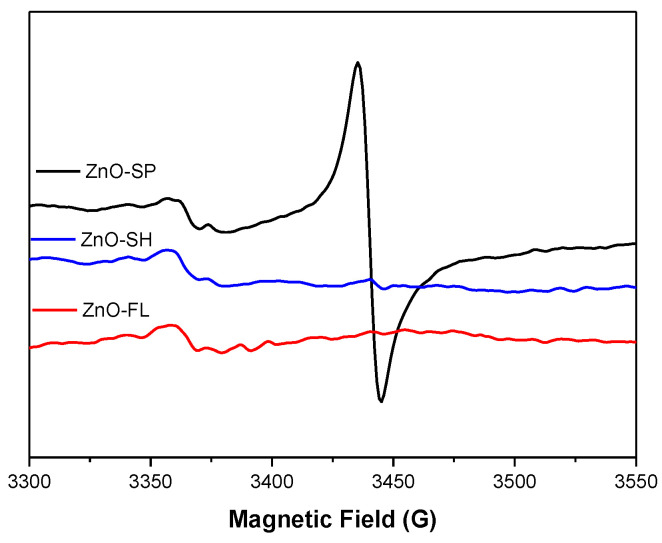
EPR spectra of the synthesized ZnO NPs in solid state at room temperature.

**Figure 8 nanomaterials-14-00638-f008:**
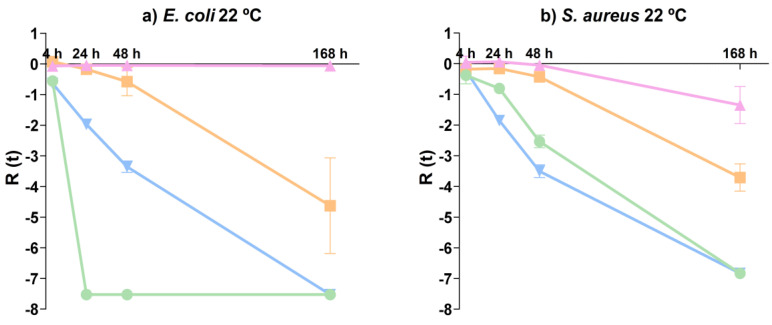
Antibacterial activity of ZnO-SP (
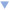
), ZnO-FL (
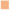
), ZnO-SH (

) and microorganism control (
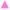
) for (**a**) *E. coli* 22 °C and (**b**) *S. aureus* 22 °C.

**Table 1 nanomaterials-14-00638-t001:** Parameters of specific surface area, pore width, and volume of ZnO-SP, ZnO-FL, and ZnO-SH.

Parameter	ZnO-SP	ZnO-FL	ZnO-SH
Specific surface area (m^2^ g^−1^)	13.4	5.3	18.5
Pore width (nm)	9.8	7.0	11.9
Pore volume (cm^3^ g^−1^)	0.03	0.02	0.07

## Data Availability

Data will be made available upon request.

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
