# Peer review of "Optimizing Antimicrobial Efficacy: Investigating the Impact of Zinc Oxide Nanoparticle Shape and Size"

_nanomaterials, 2024, doi:10.3390/nano14070638_

Round 1

Reviewer 1 Report

Comments and Suggestions for Authors

The article "Optimizing Antimicrobial Efficacy: Investigating the Impact of Zinc Oxide Nanoparticle Shape and Size" describes the synthesis, characterization and antibacterial activity of some ZnO particles with different morphology. It is a valuable study that can be published after authors address the following problems:

Abstract should be checked and revised carefully by briefly introducing the work plan and key findings.

Abstracts should highlight the innovation of the article, as often abstract section is presented separately in search engines, it must be able to stand alone as an informative piece. In the abstract, need to focus more on the quantitative information, not qualitative one. The key quantitative data showing the size, morphology, surface area, grain size and antibacterial efficiency should be included in the abstract.

Literature for the introduction of ZnO in packaging films is rather old and authors should consult, for example, works of Motelica L. et al.

Some constructions should be revised, e.g. “transparent ultraviolet light (UV) absorbers” (in abstract or row 55) as readers with no background in ZnO will not understand that authors imply that ZnO composite films can be transparent in visible light but absorb in the UV domain. If this is the original meaning… If authors have other motifs to put “UV light” between “transparent” and “absorbers” please explain.

Gram-positive/negative should be written with capital “G” as it is a person name.

Authors should evaluate the use of antibacterial and antimicrobial terms across the manuscript. While there is plenty of literature indicating that ZnO has antimicrobial activity (ok to mention it in introduction for example), when presenting own results authors should remember that they tested the nanostructures only against bacterial strains, therefore antibacterial is more suitable.

As authors used zinc oxide in title, I suggest using ZnO in keywords to improve the hit chances in search engines.

Row 112 use X-ray instead of x-ray.

Row 137 change the “zinc metal solution” with a more appropriate description: for example zinc ion solution or Zn2+ solution.

Please use proper power superscripts at row 202: 108-109 CFU/mL.

In figures 2 and 3 please change colours for ZnO-FL and ZnO-SH (red or blue or black) to improve visibility. I suggest authors to label also XRD peaks after 70o as (004) and (202) see doi: 10.3390/pharmaceutics15102470, as they declare at rows 291-292 “no additional peaks….purity of the samples” and some readers might be puzzled.

In the FTIR spectra of ZnO-SH the double peaks from 1551 and 1433 cm-1 are related to the presence of acetate ions adsorbed on surface of the nanostructures (see doi: 10.3390/pharmaceutics14122842).

To support the claim from rows 92 and 540-560 authors should look in doi: 10.3390/pharmaceutics14122842, doi: 10.3390/ijms24021629 and compare the results for the tested strains.

Figure 5 has vertical axe label unreadable.

The conclusion should reflect the heuristic of the study. How is this study advancing the knowledge? Conclusion section must be reworked to underline the novelty and advantages of this research, with actual numbers. The conclusion part does not highlight the salient findings and future perspective

Reviewer 2 Report

Comments and Suggestions for Authors

Dear editor of MDPI Nanomaterials, thank you for including me as part of your panel of reviewers for your important journal. I tell you that I have carefully reviewed the manuscript titled "Optimizing Antimicrobial Efficacy: Investigating the Impact of Zinc Oxide Nanoparticle Shape and Size." Therefore, I consider it suitable for publication after the authors make the following adjustments and respond to the comments that I include below. My decision is major revision.

Comment to authors

1. In the introduction section, the first paragraph should be separated into two paragraphs, one that talks about nanotechnology and the other about food packaging.

2. Once the first paragraph has been modified, it should include more information regarding nanotechnology. I recommend including the following information and citing the reference that I included: Nanotechnology is considered a multidisciplinary science that aids in solving current problems, and where its function is to manufacture nanoscale materials. The main applications of nanotechnology lie in the areas of food, medicine, water treatment, solar energy conversion, and catalysis. Nanotechnology is an emerging science, which is based on the design and application of nanostructures or nanomaterials that are usually in the range of 1–100 nm (nm).  (2022). Trends in sustainable green synthesis of silver nanoparticles using agri-food waste extracts and their applications in health. Journal of Nanomaterials2022, 1-37.

3. Also, once the first paragraph has been modified, you must include more information regarding food packaging. I recommend that you include the following information and cite the reference that I included: Active food packaging can be classified according to the function of the additive it contains, among which those that contain antioxidant and antimicrobial activity are the main ones, where those that possess antioxidant activity delay and prevent the appearance of free radicals, acting in oxidation reactions of the molecules present, while those that contain antimicrobial activity inhibit the action of microorganisms that cause contamination and deterioration of products. (2022). Physicochemical, structural, mechanical and antioxidant properties of zein films incorporated with no-ultrafiltered and ultrafiltered betalains extract from the beetroot (Beta vulgaris) bagasse with potential application as active food packaging. Journal of Food Engineering334, 111153.

4. It is important as a parameter to include particle size distribution and polydispersity index. The polydispersity index is obtained by dividing the standard deviation by the mean. The polydispersity index range is from 0 to 1, where close to zero is monodisperse and close to one is polydisperse.

5.  In the results section, specifically in the FT-IR section, discuss more about the differences in the spectra and give reasons why those bands are present in the spectra. In addition, you should compare with other previous works.

6. In the introduction section, specifically in adsorption desorption, you must explain what type II isotherm means in your material.

Author Response

Please see file attached

Reviewer 3 Report

Comments and Suggestions for Authors

The manuscript titled “Optimizing Antimicrobial Efficacy: Investigating the Impact of Zinc Oxide Nanoparticle Shape and Size” was aimed to synthesize ZnO NPs with spherical, flower and sheet morphologies and investigate the influence of morphology and size on antibacterial activity toward E. coli (gram-negative bacteria) and S. aureus (gram-positive bacteria). The topic is relevant. The work is well-designed and the manuscript deserves consideration. However, at presented state the manuscript needs revision. The main comments and recommendations are listed below.

1. It is better to avoid the use of abbreviation in Abstract without definition.

2. Abstract can be reached by more results (data) obtained to point out the significance and importance of the work.

3. References format should be revised in accordance with the journal guides.

4. Introduction. Discussing morphology and size of ZnO NPs, stability issue and role of stabilizing agents should be mentioned and discussed as well regarding antimicrobial potential of ZnO NPs.

5. Antimicrobial activity of ZnO NPs is well-known. To justify novelty and importance of the work, the authors could discuss in the Introduction the recent directions of application of ZnO NPs with different morphology, size and stability (or nature of stabilizing agents) associated with their enhanced antimicrobial activity. For example, wound healing preparation (https://doi.org/10.3390/gels9010057), active packaging films (https://doi.org/10.3390/ijms22041717), agricultural seed growth stimulation, food preservation,  etc.

6. Materials and Methods: details for chemicals, materials, equipment and software should be unified: (Manufacturer, City, Country), as required by MDPI

7. L. 186-188. Reference is needed

8. L. 205. Why the authors chose exact such periods? It should be justified in the text.

9. Figures 2-4 can be combined, as suggestion. In addition, in figure 3 and figure 4, line numbers moved to the figure and made a mix with axes that can confuse readers. The same for figure 6. Please revise.  

10. Figure 5. ZnO-SH graph should be revised.

11. Clarity of figure 7 should be improved, if possible.

12. Can the authors add photos with inhibition zones of E. coli and S. aureus. This is needed to assess reproducibility of the experiment.  

13. The authors cannot declare antimicrobial activity of ZnO NPs testing only 2 strains. It is recommended to use more general terms as effect on E. coli/ S. aureus or activity towards E. coli/ S. aureus. As a result, the authors can write about potential antimicrobial activity that should be tested on other strains. 

14. Discussion part should be strengthened by more references to relevant recent works with comparison of findings.  

15. Conclusions should be reached by more results (data) obtained to support each statement.

16. The text should be checked for typos and grammatical errors.

Comments on the Quality of English Language

The text should be checked for typos and grammatical errors.

Author Response

Please see file attached

Round 2

Reviewer 1 Report

Comments and Suggestions for Authors

The authors have responded to my comments and have addressed all my concerns, substantially improving the manuscript, therefore, I suggest publishing the paper titled "Optimizing Antimicrobial Efficacy: Investigating the Impact of Zinc Oxide Nanoparticle Shape and Size " in the current form.

Author Response

The authors wish thank the contribution of the reviewer for the paper improvement.

Reviewer 2 Report

Comments and Suggestions for Authors

The manuscript improved substantially and I consider it suitable for publication in its updated version. My decision is Accepted

Author Response

(The authors gave the same response as above.)

Reviewer 3 Report

Comments and Suggestions for Authors

The comments are given in the attached file

Author Response

Thank you very much for the contribution of the reviewer for the paper improvement. Issues 1 and 2 were corrected. the conclusions were revised and some data included, but trying to avoid repetition of information. We trust that the point raised by the reviewer was addressed properly.